# Melatonin/Nrf2/NLRP3 Connection in Mouse Heart Mitochondria during Aging

**DOI:** 10.3390/antiox9121187

**Published:** 2020-11-27

**Authors:** Marisol Fernández-Ortiz, Ramy K. A. Sayed, José Fernández-Martínez, Antonia Cionfrini, Paula Aranda-Martínez, Germaine Escames, Tomás de Haro, Darío Acuña-Castroviejo

**Affiliations:** 1Centro de Investigación Biomédica, Departamento de Fisiología, Facultad de Medicina, Instituto de Biotecnología, Parque Tecnológico de Ciencias de la Salud, Universidad de Granada, 18016 Granada, Spain; sol92@correo.ugr.es (M.F.-O.); ramy.kamal@vet.sohag.edu.eg (R.K.A.S.); josefermar@ugr.es (J.F.-M.); acionfrini@correo.ugr.es (A.C.); ampaula@correo.ugr.es (P.A.-M.); gescames@ugr.es (G.E.); 2Department of Anatomy and Embryology, Faculty of Veterinary Medicine, Sohag University, Sohag 82524, Egypt; 3CIBERfes, Ibs. Granada, 18016 Granada, Spain; 4UGC de Laboratorios Clínicos, Hospital Universitario San Cecilio, 18016 Granada, Spain; tomas.haro.sspa@juntadeandalucia.es

**Keywords:** melatonin, mitochondria, NLRP3 inflammasome, Nrf2, heart ultrastructure, apoptosis, mitochondrial dynamics

## Abstract

Aging is a major risk for cardiovascular diseases (CVD). Age-related disorders include oxidative stress, mitochondria dysfunction, and exacerbation of the NF-κB/NLRP3 innate immune response pathways. Some of the molecular mechanisms underlying these processes, however, remain unclear. This study tested the hypothesis that NLRP3 inflammasome plays a role in cardiac aging and melatonin is able to counteract its effects. With the aim of investigating the impact of NLRP3 inflammasome and the actions and target of melatonin in aged myocardium, we analyzed the expression of proteins implied in mitochondria dynamics, autophagy, apoptosis, Nrf2-dependent antioxidant response and mitochondria ultrastructure in heart of wild-type and NLRP3-knockout mice of 3, 12, and 24 months-old, with and without melatonin treatment. Our results showed that the absence of NLRP3 prevented age-related mitochondrial dynamic alterations in cardiac muscle with minimal effects in cardiac autophagy during aging. The deficiency of the inflammasome affected Bax/Bcl2 ratio, but not p53 or caspase 9. The Nrf2-antioxidant pathway was also unaffected by the absence of NLRP3. Furthermore, NLRP3-deficiency prevented the drop in autophagy and mice showed less mitochondrial damage than wild-type animals. Interestingly, melatonin treatment recovered mitochondrial dynamics altered by aging and had few effects on cardiac autophagy. Melatonin supplementation also had an anti-apoptotic action in addition to restoring Nrf2-antioxidant capacity and improving mitochondria ultrastructure altered by aging.

## 1. Introduction

Cardiovascular diseases (CVD) constitute the leading cause of death in the world, especially in industrialized countries [1]. Genetics, hypertension, diabetes, obesity, smoking, and physical inactivity have been identified as risk factors for these diseases [2]. However, aging is by far the major risk factor for cardiac dysfunction, since its prevalence increases dramatically in aged people. The connection between aging and these cardiac pathologies have been widely reported [3,4,5]. Cardiac aging correlates with hemodynamic and metabolic alterations together, with changes in the structure and function of cardiovascular tissues. Furthermore, the increase in reactive oxygen species (ROS) and the activation of inflammation-related pathways have also been documented [6,7,8]. Aging is characterized by an increase in oxidative damage and persistent activation of innate immunity resulting in immunosenescence. This immune dysregulation results in a state of age-associated chronic inflammation termed ‘inflammaging’, which plays an important role in the onset and progression of cardiovascular diseases, in addition to other age-related disorders [9,10,11,12].

The main components of the innate immunity include NF-kB and NLRP3 inflammasome. Focusing on the NLRP3 inflammasome, it consists of the scaffold protein NLRP3, the adaptor protein ASC and caspase-1, forming a multiprotein complex [13]. The NLRP3 inflammasome is induced upon different signs of cellular ‘danger’ and is responsible for the maturation of the NF-kB-dependent pro-inflammatory cytokines including interleukin-1β (IL-1β), potentiating the inflammatory response [14]. Some of these danger signals, such as ROS and mitochondrial DNA (mtDNA), come from impaired mitochondria during inflammation [15]. Additionally, age-related alterations in processes that maintain mitochondrial homeostasis, including fusion, fission, autophagy (mitophagy), and mitochondrial biogenesis, have been described. The resulting accumulation of dysfunctional mitochondria enhances ROS production and mtDNA release [16,17]. Another fact that could contribute to NLRP3 inflammasome activation is the reduced endogenous antioxidant defense capacity, which occurs during aging, in particular, the decline of transcription factor Nrf2 [18,19]. Thus, there seems to be a close relationship between aging, NF-kB/NLRP3 inflammasome response, cardiac and mitochondrial dysfunction, ROS formation, and decrease in Nrf2.

Melatonin (N-acetyl-5-methoxytryptamine, aMT) is an ubiquitous molecule that, aside from the pineal gland [20], is synthesized by most body organs and tissues, including the heart [21,22]. In addition to its chronobiotic effects, this indoleamine presents important anti-oxidative and anti-inflammatory properties that depend on the high levels of extrapineal melatonin [23,24,25]. Within the cell, melatonin acts on its main target, the mitochondria, boosting their bioenergetic properties, enhancing the ATP levels and reducing the formation of free radicals [26,27,28,29,30]. In multiple experimental conditions including acute and chronic inflammation, and aging in mouse heart, melatonin consistently prevented oxidative stress, reduced the innate immunity activation, and boosted cardiac mitochondria function [12,23,25,31].

The mechanisms by which NLRP3 contributes to cardiovascular disorders are still unclear [32]. We hypothesized that NLPR3 inflammasome has a role in aged cardiac muscle and we considered it worthwhile to evaluate its association with molecular mechanisms underlying the development of cardiovascular diseases with age. Moreover, we also hypothesized that melatonin is able to counteract the age-related changes in the myocardium and we investigated where it exerts its action. For this purpose, we assessed age-associated disturbances regarding mitochondrial dynamics (fusion/fission), autophagy (mitophagy), apoptosis, Nrf2-dependent antioxidant response, and mitochondrial ultrastructure in the heart of the wild-type and NLRP3-knockout mice at 3, 12, and 24 months of age, with and without melatonin treatment.

## 2. Materials and Methods

### 2.1. Animals and Treatment

Wild-type C57BL/6J and NLRP3-knockout mice NLRP3^−/−^ (B6.129S6-NLRP3tm1Bhk/J) on the wild-type C57BL/6J background (>10 backcrosses) aged 3 weeks were purchased from Charles River (Barcelona, Spain) and The Jackson Laboratory (Bar Harbor, ME, USA), respectively. Mice were housed in the animal facility of the University of Granada under a specific pathogen-free barrier and were kept under controlled temperature (22 °C ± 1 °C). Room illumination was on automated 12 h light/dark cycle (lights on at 08:00 h). Animals had ad libitum access to tap water and pelleted rodent chow.

This study was carried out in accordance with the National Institutes of Health Guide for the Care and Use of Laboratory Animals (National Research Council, National Academy of Sciences, Bethesda, MD, USA), the European Convention for the Protection of Vertebrate Animals used for Experimental and Other Scientific Purposes (CETS #123), and the Spanish law for animal experimentation (R.D. 53/2013). The protocol was approved by the Andalusian’s Ethical Committee (05/07/2016/130).

Wild-type (WT) and NLRP3^−/−^ mice were divided into five experimental groups (*n* = 7 animals per group) (Figure 1): (I) young (Y, 3-months old), (II) early-aged (EA, 12-months old), (III) early-aged plus melatonin (EA + aMT), (IV) old-aged (OA, 24-months old), and (V) old-aged plus melatonin (OA + aMT) mice. Melatonin (aMT) was orally administered at 10 mg/kg/day in the chow during the last two months before early and old-aged treated mice were sacrificed (EA + aMT at the age of 10 months and OA + aMT at the age of 22 months). The other groups of animals (Y, EA and OA) were fed with normal chow without melatonin. The melatonin pelleted chow was prepared by the Diet Production Unit facility of the University of Granada. The amount of melatonin in the pellets was calculated according to the average daily food intake, number, weight and age of mice [33]. The use of 10 mg/kg/day was selected on the basis of previous studies that demonstrated the effectiveness of this dose on the aging process [11,34] and mitochondrial function [35,36]. C57/BL6A was reported to be a strain of mice that responds well to melatonin therapy [37,38]. Therefore, we deemed them suitable for the purpose of this study.

Animals were killed by cervical dislocation after ketamine plus xylazine anesthesia, and hearts were collected. The left ventricle was dissected and divided into two parts. One part was washed in saline, and rapidly fixed in 2.5% glutaraldehyde for transmission electron microscopy analysis, while the other part was stored at −80 °C for further western blot analysis.

### 2.2. Western Blot Analysis

Pure cytosolic subcellular fraction was isolated from heart tissue according to Dimauro et al. [39] with some adjustments described in Rahim et al. [30]. Briefly, heart tissue was homogenized on ice at 800 rpm in 500 μL of STM buffer containing 250 mM sucrose, 50 mM Tris-HCl pH 7.4, 5 mM MgCl_2_, 0.5 mM DTT, 5% phosphatase inhibitor buffer (125 mM NaF, 250 mM β-glycerophosphate, 250 mM p-nitrophenyl phosphate, and 25 mM NaVO3), and a protease inhibitor cocktail (Cat. 78429, Thermo Fisher Scientific, Waltham, MA, USA) with a Teflon pestle. The homogenate was maintained on ice for 30 min, then centrifuged at 800 g for 15 min at 4 °C. The supernatant was labeled as S0 and used for subsequent isolation of cytosolic fractions. S0 was centrifuged at 800× *g* for 10 min at 4 °C and the supernatant S1 was centrifuged at 11,000× *g* for 10 min. The resulting supernatant S2, containing cytosol and microsomal fraction, was precipitated in cold 100% acetone at −20 °C for 1 h followed by centrifugation at 12,000× *g* for 5 min. The pellet was then resuspended in 300 μL STM buffer and labeled as cytosolic fraction.

Western blot analysis was performed on cytosolic fractions of mice hearts. Denatured protein samples (40 μg/fraction) were separated by sodium dodecyl sulfate polyacrylamide gel electrophoresis (SDS-PAGE) using 12% or 15% acrylamide/bis-acrylamide gels. Proteins were then wet transferred to a polyvinylidene difluoride (PVDF) membrane (Merck Life Science S.L.U., Madrid, Spain). The membrane was blocked in 5% bovine serum albumin (BSA) in PBST (PBS with 0.1% Tween-20) at room temperature and then incubated overnight at 4ºC with the primary antibodies (Appendix A) diluted in blocking buffer as per manufacturer’s specification. Membranes were washed with PBST 3 × 10 min and incubated for 1 h at room temperature with anti-mouse (BD Biosciences Pharmigen, San Jose, CA, USA) or anti-rabbit (Thermo Scientific, Madrid, Spain) IgG-horseradish peroxidase conjugated secondary antibodies diluted according to the manufacturer’s instructions. After washing with PBST, immunoreaction was detected using ClarityTM Western ECL Substrate (Bio-Rad, Madrid, Spain) and revealed in Kodak Image Station 4000MM PRO (Carestream Health, Rochester, NY, USA). Bands were analyzed and quantified using Kodak Molecular Imaging Software v. 4.5.1 (Carestream Health, Rochester, NY, USA). GAPDH protein content was used to normalize the cytosolic subcellular fraction. Data obtained from early and old-aged mice were always compared to young mice of the same group. The value of WT Y mice group was defined as 100%. Full images of western blots are available in Appendix A.

### 2.3. Transmission Electron Microscopy (TEM)

Small pieces from the left ventricle of the heart were rapidly immersed in a 2.5% glutaraldehyde in 0.1M cacodylate buffer (pH 7.4) for fixation, then post-fixed in 0.1 M cacodylate buffer-with 1% osmium tetroxide and 1% potassium ferrocyanide for 1 h. The specimens were then immersed in 0.15% tannic acid for 50 s, incubated in 1% uranyl acetate for 1.5 h, dehydrated in ethanol, and embedded in resin. Ultrathin sections of 65 nm thickness were cut using a Reichert-Jung Ultracut E ultramicrotome. These sections were double stained with uranyl acetate and lead citrate [40], and examined by a Carl Zeiss Leo 906E electron microscope and digital electron micrographs were acquired.

### 2.4. Morphometric Analyses

Using electron micrographs, mitochondrial number and percentage of the mitochondrial damage (as number of damaged mitochondria/total mitochondrial number 100) were analyzed in areas with a width and height of 5.24 µm and 3.99 µm, respectively. Moreover, some morphometric analyses, including cross-sectional area (CSA) and Feret’s diameter of the intermyofibrillar mitochondria of cardiac muscle fibers were performed on images of electron microscopy using Image J processing software.

### 2.5. Statistical Analyses

Data are expressed as mean ± standard error of the mean (SEM) of *n* = 7 animals per group. All statistical analyses were carried out using GraphPad Prism 6.0 software (GraphPad Software, San Diego, CA, USA). One-way ANOVA with a Tukey’s post hoc test was used to compare the differences between experimental groups. The values were found to be significantly different when *p* < 0.05.

## 3. Results

### 3.1. NLRP3 Deficiency Prevents, and Melatonin Treatment Restores Cardiac Muscle Mitochondrial Dynamics Altered by Aging

Anomalies in mitochondrial dynamics (fusion/fission) are typical of aged cardiac muscle [16]. Here, we showed that aging induced a decrease in the levels of proteins involved in mitochondrial dynamics, including Mfn2, Opa1, and Drp1, in WT mice, an effect absent in NLRP3^−/−^ mice (Figure 2A–C). Melatonin supplementation counteracted the decline of Mfn2, Opa1, and Drp1 caused by aging in WT mice. Interestingly, no significant effect of melatonin was observed in fusion proteins Mfn2 and Opa1 in NLRP3^−/−^ mice at the age of 12 and 24 months (Figure 2A,B). A slight, but not significant enhancement in fission protein Drp1 was noted in EA and OA NLRP3^−/−^ mice with melatonin supplementation (Figure 2C).

### 3.2. NLRP3 Deficiency and Melatonin Therapy Had Minimal Effects in Autophagy in Cardiac Muscle during Aging

A drop in the autophagic capacity observed in cardiac aging is associated with the accumulation of dysfunctional mitochondria, exaggerated ROS production, and mtDNA release [16,17]. Unsurprisingly, the conversion of LC3I to LC3II, a hallmark of autophagy [41], was significantly reduced in WT mice during aging, as reflected in the decrease in the LC3II/LC3I ratio in WT EA and OA mice (Figure 3). LC3II/LC3I ratio trends to increase in NLRP3^−/−^ EA and OA mice, which may explain the attempt to restore autophagy events. Melatonin administration had minimal effects on the LC3II/LC3I ratio in all cases.

### 3.3. Melatonin Treatment and, to a Lesser Extent NLRP3 Deficiency, Reduced Apoptosis in Cardiac Muscle during Aging

Despite being intensively studied over the past three decades, many of the mechanisms of apoptotic cell death remain unknown. Although the relationship between aging and apoptosis have been a subject of controversy in scientific community, there seems to be consensus that apoptosis plays a significant role in cardiac aging [42]. Here, we showed that aging induced a rise in the levels of some proteins involved in apoptotic processes, including p53 and caspase 9 in both WT and NLRP3^−/−^ mice. Melatonin treatment significantly diminished the levels of p53 and caspase 9 in EA WT mice and in EA and OA mutant mice (Figure 4A,B). The pro-apoptotic protein Bax and the anti-apoptotic Bcl2 were significantly enhanced by aging in WT mice. Mutant mice only showed Bcl2 increased in OA animal’s group (Figure 4C,D). We observed a slight rise in Bax/Bcl2 ratio in EA and a significantly increase in WT OA mice (Figure 4E). The absence of NLRP3, however, prevented the apoptotic process associated with aging since Bax/Bcl2 ratio remained at similar levels as that of Y mutant mice. Melatonin supplementation significantly decreased the Bax/Bcl2 ratio in EA and OA WT mice, but had no effect in NLRP3^−/−^ mice.

### 3.4. Melatonin Treatment, but not NLRP3 Deficiency, Recovered the Nrf2-Dependent Antioxidant Capacity in Cardiac Muscle Declined by Aging

In recent years, emerging evidence has indicated that aging leads to a gradual reduction of the Nrf2-dependent antioxidant response, which in turn contributes to the accumulation of oxidative stress [18,19]. Our results showed a significant decrease in the protein levels of Nrf2 and its active form pNrf2 (Ser40) in WT and NLRP3^−/−^ mice with age, suggesting that NLRP3 deficiency was unable to ameliorate the age-related decline of Nrf2 and pNrf2 (Ser40) in these animals (Figure 5A,B). Melatonin supplementation markedly recovered the levels of Nrf2 and pNrf2 (Ser40) in both WT and mutant EA and OA mice. Aging and melatonin therapy did not significantly modify the levels of the Nrf2 inhibitor, Keap1, in either mouse strain (Figure 5C). Hmox1, Nqo1, and γGclc, three cytoprotective enzymes transcriptionally regulated by Nrf2, also remarkably decreased in WT OA mice (Figure 5D–F). The levels of Hmox1 and γGclc significantly dropped in NLRP3^−/−^ EA and OA mice (Figure 5D,F). Protein content of Nqo1 enzyme was not modified by aging in mutant animals (Figure 5E). Again, melatonin treatment greatly enhanced the levels of Hmox1, Nqo1, and γGclc in WT and NLRP3^−/−^ mice.

### 3.5. NLRP3 Deficiency and Melatonin Therapy Improved Mitochondria Ultrastructure Altered by Age in Cardiac Muscle

Transmission electron microscopy of the cardiac muscles of Y WT mice revealed the presence of normally intact and compacted mitochondria with clearly organized cristae distributed in the intermyofibrillar spaces (Figure 6A,B). At the age of 12 months (EA), most of these mitochondria were found normally; however, a few showed cristae damage (Figure 6C,D). These changes were exacerbated and numerous mitochondria were severely damaged, hypertrophied, and vacuolated with completely destroyed cristae in WT OA mice (Figure 6G,H). Melatonin supplementation, however, preserved the normal ultrastructure of the cardiac mitochondria in EA (Figure 6E,F) and OA WT mice (Figure 6I,J), maintaining their healthy and compact appearance.

Cardiac muscle fibers of NLRP3^−/−^ Y mice presented normal highly compacted mitochondria with densely packed cristae (Figure 7A,B). Mitochondrial structure did not change in EA mice, except one that showed damage in peripheral cristae (Figure 7C,D). The mitochondrial damage was less prevalent at 24 months in comparison with WT OA mice. Mitochondria were characterized by their widely-separated and organized cristae, with the presence of small-sized membranous vacuoles of possibly autophagic nature (Figure 7G,H). Melatonin treatment exhibited an obvious protective effect at the age of 12 (Figure 7E,F) and 24 months (Figure 7I,J), where it kept normal mitochondrial architecture with aging, in addition to formation of multivesicular bodies, which reflect the induction of the autophagic processes.

### 3.6. Lack of NLRP3 Reduced Mitochondria Number Loss and Mitochondrial Damage, an Effect Shared by Melatonin

Morphometric analysis of cardiac mitochondria revealed that mitochondrial number exhibited initial non-significant decline in cardiac muscles of WT and NLRP3^−/−^ EA mice. Nevertheless, mitochondrial number was significantly decreased in OA, being more pronounced in WT mice than NLRP3^−/−^ one, an effect significantly counteracted after melatonin therapy (Figure 8A). Furthermore, the percentage of the mitochondrial damage was significantly increased in aged mice, especially in WT animals, and it was counteracted by melatonin supplementation (Figure 8B). Morphometrical analysis of the mitochondrial CSA illustrated a non-significant increase in cardiac muscle of WT and NLRP3^−/−^ EA mice, whereas the former increased in aged animals (Figure 8C). Mitochondrial diameter showed non-significant increase in WT EA mice, increasing in OA animals. NLRP3^−/−^ mice revealed non-significant changes in mitochondrial diameter among all experimental groups (Figure 8D).

## 4. Discussion

Immunosenescence and inflammaging are caused by persistent activation of NF-κB/NLRP3 inflammasome pathways generates chronic low-grade inflammation, which leads to, among other detriments, accumulation of cardiac mitochondrial dysfunction, characterized by dysregulation of mitochondrial dynamics, autophagy, apoptosis, Nrf2 antioxidant pathway, and maintenance of ultrastructure of mitochondria [43]. Another hallmark of aging is a decline in melatonin levels and its protective roles [44]. This brings about increased oxidative damage, chronodisruption, upregulation of pro-inflammatory cytokines, and downregulation of anti-oxidant/-inflammatory processes that contribute to inflammaging by facilitating mitochondrial disruption [45]. The role of the NLRP3 inflammasome and melatonin levels in regulation of mitochondrial dysfunction, associated with cardiac aging, is not fully understood. Our results suggest direct involvement of this inflammasome by marked amelioration of some mitochondrial dysfunctions with NLRP3 ablation both involved with, and independent of, melatonin supplementation in EA and OA mice (Figure 9 and Figure 10).

Mitochondria fusion (Mfn2 and Opa1) and fission (Drp1) proteins decrease naturally with aging, as seen in WT mice (Figure 9A). Findings in the literature link declines in regulatory proteins of mitochondrial dynamics and age-related development of cardiovascular disease [46,47,48,49]. Cardiomyocytes of Mfn2-deficient mice showed cardiac hypertrophy [50]. Low levels of Opa1 have been reported in failing human heart [51]. Loss of Drp1 in adult mice results in lethal dilated cardiomyopathy [52]. Our study also concluded that the absence of NLRP3 prevented the decrease in fusion and fission processes associated with aging that were observed in WT mice (Figure 10A). This cardioprotective effect observed in NLRP3^−/−^ mice supports the existence of a close relationship between mitochondrial dynamics and inflammaging. Our results are in line with scientific evidence that connects impaired mitochondria dynamics, stimulation of innate immune response and inflammasome activation [53,54,55,56,57,58]. On the other hand, melatonin’s mechanism of action in mitochondria dynamics and aging remains unclear. We indicate herein that melatonin promotes fusion by increasing the expression of the Mfn2 and Opa1 proteins in WT EA and OA mice (Figure 9A). Most investigations agree that this indolamine stimulates mitochondria fusion, contributing to the survival of cardiomyocytes and reducing mitochondria damage [59,60,61]. Moreover, numerous studies remark a melatonin-induced reduction of mitochondria fission with stressful stimuli [62,63,64,65], showing a protective effect in cardiac function against ischemia/reperfusion injury and post-traumatic cardiac dysfunction in vitro and in vivo models, respectively [66,67,68]. Conversely, we found that melatonin supplementation increased the levels of the Drp1 protein in EA and OA WT mice. Supporting our results, recent findings showed that increasement in Drp1 levels enhanced regulation of mitochondria homeostasis through mitophagy [69]. Additionally, Drp1 overexpression in flies reversed age-related mitochondria dysfunction and age-onset pathologies [70]. Taken together, our data suggests that melatonin enhances the response of mitochondria dynamics to maintain homeostasis during age-related metabolic stressors like inflammasome activation. It should be noted that melatonin did not trigger significant changes in EA and OA NLRP3^−/−^ mice either (Figure 10A). This effect of melatonin has previously been related to its cytoprotective activity, since its effect will be greater the more cellular damage there is, while in situations of low damage or physiological conditions its response is minimal [71].

The LC3II/LC3I ratio showed a significant decrease in autophagy in EA and OA WT mice compared to Y WT mice (Figure 9B). Numerous findings indicate a loss of autophagy with aging in most organisms and tissues, including the heart [72,73,74,75,76]. Changes in the expression of autophagic proteins such as Atg9, LAMP-1, and LC3II in aged mice and rats resulted in cardiac dysfunction [77,78,79]. The consequent accumulation of altered organelles, mutated mtDNA, cristae disarray, and ROS, have been shown to propagate different age-related cardiac pathologies [74,75,80,81,82] and produce risk-associated molecular pattern derived from mitochondria (DAMP) that activate NLRP3 inflammasome [83]. Our results showed that the absence of NLRP3 prevented the drop in LC3II/LC3I ratio in mice during aging (Figure 10B). Ablation of the NLRP3 inflammasome in old NLRP3^−/−^ mice has been reported to improve the quality of autophagy by increasing the levels of ATG12, beclin 1 and LC3II and decreasing p62/SQSTM1 [84]. Several studies have demonstrated the protective influence of melatonin by both increasing and decreasing autophagic capacity, in response to sterile and non-sterile inflammation [85,86,87,88,89,90,91,92]. Interestingly, in our results, it is implied that melatonin had no effect on EA and OA WT mice compared to their corresponding controls (Figure 9B). Similar results were obtained in the brain of SAMP8 mice, where melatonin did not cause changes in autophagy [93]. However, it is noteworthy that melatonin was able to increase autophagy of OA WT mice, thereby restoring levels like Y WT mice, but not in EA mice. This action suggests that melatonin and autophagy operate synergistically to increase cell survival, delay immunosenescence, and decrease oxidative stress. Thus, melatonin could act selectively, increasing autophagy only when antioxidant activity is severely impaired, or when sufficient loss of cellular homeostasis results in abnormal mitochondrial morphology and death receptor pathway activation [94,95,96,97,98]. Melatonin did not cause significant changes in the LC3II/LC3I ratio in NLRP3^−/−^ mice (Figure 10B), possibly due to the protective effect resulting from ablation of the inflammasome.

Apoptotic proteins p53 and caspase 9 were found to be increased in EA and OA vs. Y mice in both WT and NLP3^−/−^ mice (Figure 9C,D and Figure 10C,D). Oxidative stress that occurs during aging has been shown to induce apoptosis, mitochondria dysfunction in cardiomyocytes, and ultimately heart failure [99,100,101,102]. The Bax/Bcl2 ratio confirmed the increase in apoptosis with aging in WT mice. Interestingly, no changes were observed between ages in mutant mice. The ablation of NLRP3 had an anti-apoptotic protective effect during cardiac aging in Bax/Bcl2 ratio, but not in p53 or caspase 9. This finding suggests that NLRP3 is a direct regulator of the intrinsic apoptotic pathway in cardiac aging, which is dependent on the balance between Bax and Bcl2 and cytochrome c release (Figure 10C). The absence of this inflammasome could trigger activation of extrinsic apoptosis with ligand-induced activation of several death receptors since the participation of p53 and caspase 9 in this pathway has been reported in various tissues and cell models [103,104,105]. In support of our hypothesis, recent studies revealed an increase in TNFα in the serum of old NLRP3^−/−^ mice compared to young mice [106]. This cytokine is linked to inflammaging [107] and induces extrinsic apoptotic pathway by binding to the cell death receptor TNFR1. On the other hand, findings have showed that caspase 8, which is key in extrinsic apoptosis, plays a role in NLRP3 inflammasome priming and cytochrome c independent caspase 9 activation [108,109,110]. Without NLRP3, cardiac aging-induced inflammation is favored and could start with extrinsic TNFα apoptosis pathway preceding activation of caspase 8, which in turn activates caspase 9 (Figure 10D). Further investigations centered on the impact of aging on the heart are required to elucidate the extent of the complex interactions between NLRP3 and apoptosis. In most cases, melatonin counteracted the high levels of p53 and caspase 9 associated with aging in WT and mutant mice and Bax/Bcl2 ratio in WT. This anti-apoptotic effect of melatonin during cardiac aging was evident in both extrinsic and intrinsic pathways (Figure 9C,D and Figure 10C,D) and can be explained due to its ability to restore the redox potential of the mitochondria membrane and reduce oxidative stress. These actions increase ATP production and decrease mitochondrial outer membrane permeabilization (MOMP) following release of cytochrome c [111].

Mitochondrial theory of aging [112,113] postulates that an alteration in the redox state of the mitochondria, the main source of free radicals in the cell, causes oxidative damage that results in senescence, the primary driver of the aging process. In this sense, Nrf2 is defined as a ‘guardian of health span’ and a ‘master regulator of aging’ giving it enormous importance in the control of numerous antioxidant enzymes [114,115]. It is well stablished that Nrf2 improves mitochondria function by balancing reduction and oxidation processes and influencing ATP production, membrane potential, fatty acid oxidation, and structural integrity [116]. However, changes in the levels of this protein during aging, as well as the antioxidant enzymes it regulates, have been the subject of debate in recent years. Controversial and even opposite results appear in many studies, which seem to depend on the species, strain, tissue, sex and experimental design. Our results in cardiac muscle indicate that cytosolic levels of Nrf2 and pNrf2 (Ser40) decrease with aging, both in WT and in NLRP3^−/−^ mice at EA and OA (Figure 9E and Figure 10E). This may suggest translocation to the nucleus to activate transcription, to mediate age-related increases in ROS, decreasing cytosolic levels. The presence of pNrf2 in the cytosol could also be due to phosphorylation of Nrf2 by GSK-3β which translocates pNrf2 out of the nucleus [117]. Our data agree with investigations showing that mice deficient in Nrf2 have a higher susceptibility to inflammation and oxidative stress [118]. This alteration in the Nrf2 pathway is associated with cardiovascular diseases [119,120]. Nrf2^−/−^ mice were more prone to heart failure and their mortality increased ten days after suffering a myocardial infarction [121,122]. Although most studies point to a decrease in Nrf2 in heart tissue with aging, the causes are unknown. Surprisingly, our results discarded Keap1 as the responsible of this declining since there were no changes in its levels between the different ages and experimental groups. In line with our findings, levels of Nrf2 and its mRNA were found to be reduced in the liver of 10-month-old SAMP8 mice compared to SAMR1 mice, while Keap1 mRNA and its protein levels remained unchanged with age [117]. The decrease in the antioxidant enzymes Hmox1, Nqo1, and γGclc during aging is possibly due to a less efficient Nrf2 signaling [123,124]. Similar results using aortas of 24-month-old rats, whose Nrf2 levels were lower compared to 3-month-old young rats, resulted in a drop in the enzymes Hmox1, Nqo1 and γGclc [125]. However, the same group demonstrated that oxidative stress associated with aging did not induce significant changes in Nrf2 levels of carotid arteries in aged Rhesus macaques (20 years) compared to young individuals (10 years), and their respective antioxidant enzymes were not induced either [126]. Together, these data confirm that the expression of these antioxidant enzymes is linked to Nrf2. It also suggests the activation of this signaling pathway in the cardiovascular system during aging depends not only on the animal model but on the degree of oxidative stress as well. In this light, recent works described that there is a shift in Nrf2 target to Klf9 instead of Hmox1, Nqo1, and γGclc at excessive oxidative damage [127,128]. This could explain the fact that Hmox1 and γGclc were decreased to a greater degree than WT by showing decline in EA while WT decreased only at OA. Interestingly, Nqo1 expression levels were not reduced in NLRP3^−/−^ mice but were still upregulated by melatonin supplementation. Several studies show that Nqo1 is the prototype gene target for Nrf2 activation. In BV2 cells after cerebral ischemia reperfusion, Nrf2 ROS response was linked to Nqo1 expression [129]. This could illuminate the limited decrease in cytosolic Nqo1 by being preferentially targeted by the ever-shrinking pool of Nrf2 and pNrf2 as aging ensues. This study also proved that scavenging of ROS by Nqo1 restrained NLRP3 inflammasome activation and IL-1β expression. Except for Keap1 expression levels, which remained unchanged during aging, treatment with melatonin counteracted the age-associated decline in expression of all the parameters of the Nrf2 signaling pathway, both in WT and NLRP3^−/−^ mice (Figure 9E and Figure 10E). Melatonin has been shown to have a protective effect on the mitochondria by acting as a powerful antioxidant in a direct way, as a scavenger of free radicals, detoxifying reactive oxygen and nitrogen species, and indirectly, increasing the rest of the Nrf2-dependent and independent antioxidant systems [130,131,132,133].

Studies in animal models confirm that the ultrastructure of cardiac mitochondria changes with aging [134]. Our study supported these results. A small number of isolated mitochondria had damaged cristae in EA mice, and severe mitochondrial damage, with destroyed, separated, vacuolated and hypertrophied cristae in OA mice. This mitochondrial impairment was more remarkable in WT mice than in mutants. These findings reveal age-induced cellular senescence and mitochondrial dysfunction [135], as well as the cardioprotective effect linked to the ablation of the NLRP3 inflammasome [32]. Melatonin treatment maintained normal mitochondrial ultrastructure in all experimental groups. Multivesicular bodies increased in treated OA NLRP3^−/−^ mice, which indicate autophagy induction. These results, once again, highlight the protective role of melatonin against age-mediated mitochondria impairment and its ability to restore altered autophagic processes during cardiac aging [34].

Various morphometric analyses show that the size and number of mitochondria per cell is impacted during cardiac aging [136]. Our results showed an increase in CSA and Feret’s diameter in the mitochondria of OA WT mice, accompanied by a decrease in number of mitochondria. This mitochondrial hypertrophy has been related to a systemic demand from overload stress on the heart [137,138], and our results suggest that it could also be an adaptive mechanism to compensate for the decrease in the amount of this organelle. The ablation of the NLRP3 inflammasome reduced cardiac hypertrophy, as there were no changes in Feret’s diameter with age and less significant increase in CSA and decline in mitochondria number. To our knowledge, this is the first time that these morphometrical parameters are studied specifically in IMF during cardiac aging using a mice model. In line with our findings, CSA of cardiomyocytes from the left ventricle of male Fischer 344 rats increased with aging, while the number of cardiomyocytes decreased [139]. In Wistar rats, mitochondria volume fraction and mean size both in left and right ventricle were decreased in 2 years old vs. 6 weeks old animals [140]. Our results showed that melatonin significantly increased the number of mitochondria in WT and NLRP3^−/−^ mice, with no effect on CSA or Feret’s diameter. It is possible that in this case, two-month treatment is not enough to counter the age-related changes in CSA and Feret’s diameter in the heart, one of the most energy-demanding organs of our body [141]. This ‘cardiac sarcopenia’ has hardly been investigated since most studies focus on skeletal muscle. Indeed, our group previously performed the same analyses in gastrocnemius and morphometric alterations were observed earlier, in EA mice and protected in NLRP3 deficient mice [142]. Our findings suggest that cardiac muscle and its mitochondria are physiologically more protected from age-related sarcopenia than skeletal muscle. Its ability to make a metabolic switch in favor of glycolysis instead of fatty acid oxidation during aging [143,144], being one of the organs where the NLRP3 inflammasome is expressed less [145,146], or the presence of resident macrophages with tissue protective function [147] are some of the many possible adaptations of the heart that could explain its greater resistance to sarcopenia.

## 5. Conclusions

The results of this study clarify the impact of NLRP3 inflammasome and melatonin treatment in the mitochondria during cardiac aging. The main findings can be summarized as (1) NLRP3 knocking out and melatonin supplementation avoided mitochondrial dynamics changes of heart with aging; (2) loss of NLRP3 and melatonin treatment revealed few impacts on cardiac autophagy during aging; (3) NLRP3 absence had a less role on cardiac apoptosis during aging compared to melatonin therapy; (4) melatonin restored aged-related Nrf2-dependent antioxidant capacity while NLRP3 inflammasome showed no effect on this pathway; (5) lack of NLRP3 as well as melatonin treatment enhanced mitochondria ultrastructure alterations in aged myocardium.

## Figures and Tables

**Figure 1 antioxidants-09-01187-f001:**
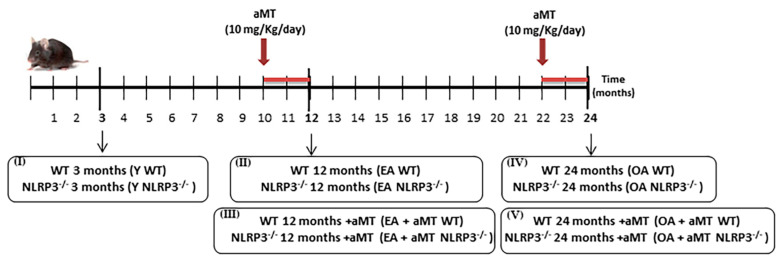
Study design summary: experimental groups and melatonin treatment.

**Figure 2 antioxidants-09-01187-f002:**
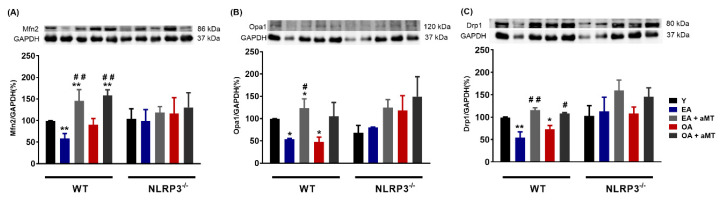
Changes in mitochondrial dynamics (fusion/fission) in WT and NLRP3^−/−^ mice during aging and melatonin treatment. (**A**) Protein levels of Mfn2. (**B**) Protein levels of Opa1. (**C**) Protein levels of Drp1. Experiments were performed in hearts of young (Y), early-aged (EA), early-aged with melatonin (EA + aMT), old-aged (OA), and old-aged with melatonin (OA + aMT) wild type and NLRP3^−/−^ mice. Data are expressed as means ± SEM (*n* = 7 animals/group). * *p* < 0.05, ** *p* < 0.01 vs. Y; ^#^
*p* < 0.05, ^##^
*p* < 0.01 vs. group without melatonin treatment.

**Figure 3 antioxidants-09-01187-f003:**
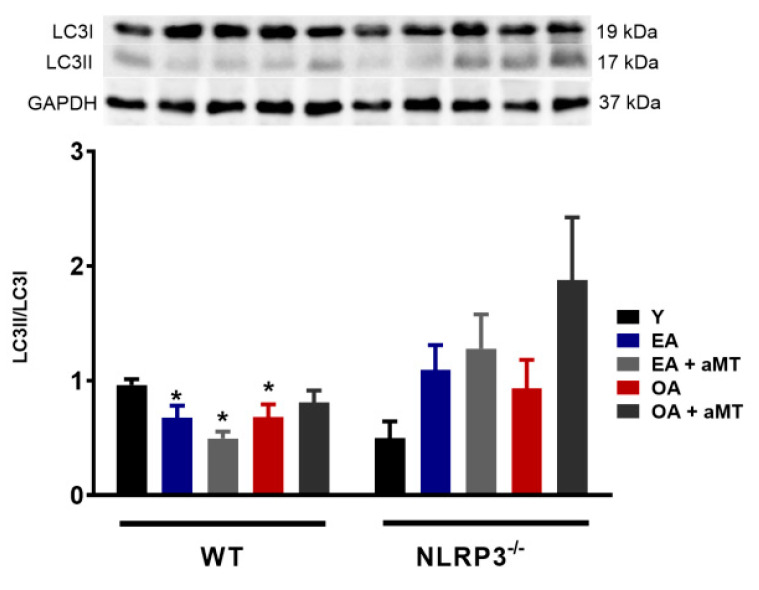
Changes in autophagy in WT and NLRP3^−/−^ mice during aging and melatonin treatment. LC3II/LC3I ratio. Experiments were performed in hearts of young (Y), early-aged (EA), early-aged with melatonin (EA + aMT), old-aged (OA), and old-aged with melatonin (OA + aMT) wild type and NLRP3^−/−^ mice. Data are expressed as means ± SEM (*n* = 7 animals/group). * *p* < 0.05 vs. Y.

**Figure 4 antioxidants-09-01187-f004:**
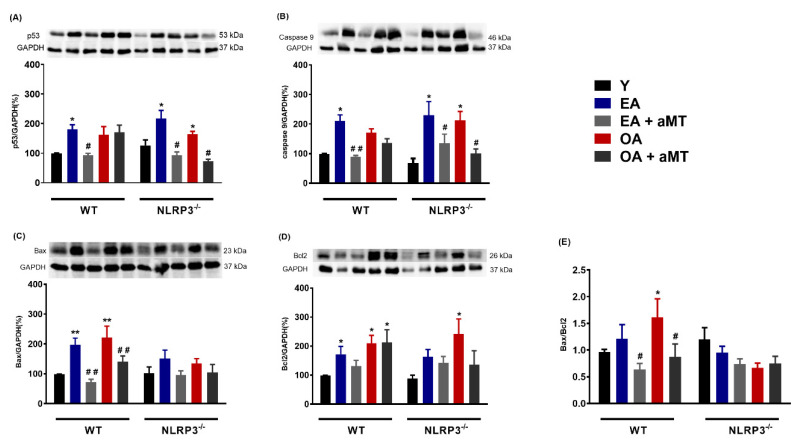
Changes in apoptosis in WT and NLRP3^−/−^ mice during aging and melatonin treatment. (**A**) Protein levels of p53. (**B**) Protein levels of caspase 9. (**C**) Protein levels of Bax. (**D**) Protein levels of Bcl2. (**E**) Bax/Bcl2 ratio. Experiments were performed in hearts of young (Y), early-aged (EA), early-aged with melatonin (EA + aMT), old-aged (OA), and old-aged with melatonin (OA + aMT) wild type and NLRP3^−/−^ mice. Data are expressed as means ± SEM (*n* = 7 animals/group). * *p* < 0.05, ** *p* < 0.01 vs. Y; ^#^
*p* < 0.05, ^##^
*p* < 0.01 vs. group without melatonin treatment.

**Figure 5 antioxidants-09-01187-f005:**
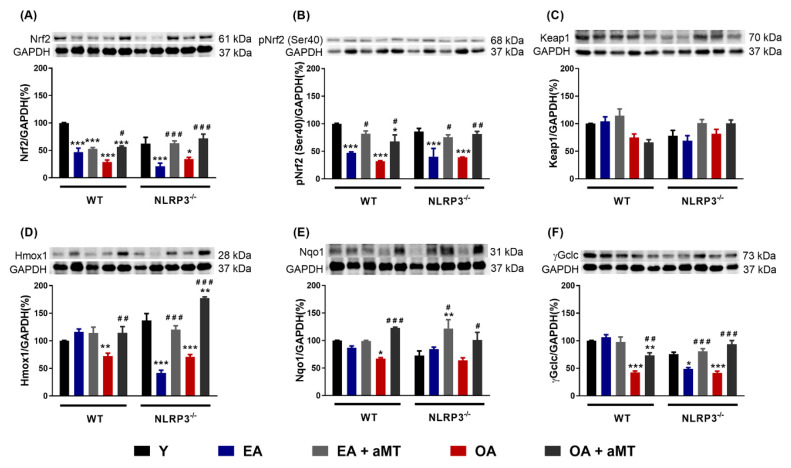
Changes in the Nrf2-dependent antioxidant pathway in WT and NLRP3^−/−^ mice during aging and melatonin treatment. (**A**) Protein levels of Nrf2. (**B**) Protein levels of pNrf2 (Ser40). (**C**) Protein levels of Keap1. (**D**) Protein levels of Hmox1. (**E**) Protein levels of Nqo1. (**F**) Protein levels of γGclc. Experiments were performed in hearts of young (Y), early-aged (EA), early-aged with melatonin (EA + aMT), old-aged (OA), and old-aged with melatonin (OA + aMT) wild type and NLRP3^−/−^ mice. Data are expressed as means ± SEM (*n* = 7 animals/group). * *p* < 0.05, ** *p* < 0.01, *** *p* < 0.001 vs. Y; *^#^ p* < 0.05, ^##^
*p* < 0.01, ^###^
*p* < 0.001 vs. group without melatonin treatment.

**Figure 6 antioxidants-09-01187-f006:**
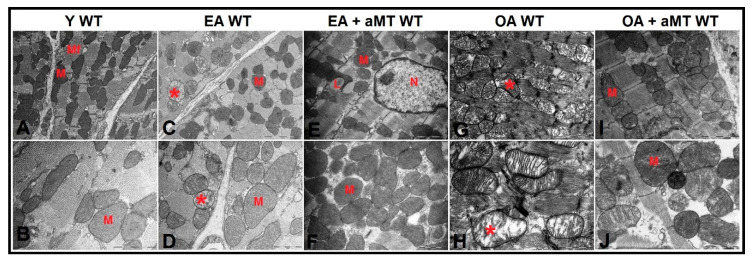
Age-associated ultrastructural changes of mitochondria in cardiac muscle fibers of WT mice and melatonin treatment. (**A**,**B**) Electron micrographs of cardiac muscle fibers of Y WT mice revealing the presence of normally intact and compacted mitochondria (M) distributed among myofibrils (Mf). (**C**,**D**) Electron micrographs of cardiac muscle fibers of EA WT mice demonstrating the presence of normal mitochondria (M) with few demonstrating cristae damage (asterisk). (**E**,**F**) Electron micrographs of cardiac muscle fibers of EA + aMT WT mice showing the protective effect of melatonin supplementation in preserving normal mitochondrial structure (M) with the presence of lipid droplets (L), N; nucleus. (**G**,**H**) Electron micrographs of cardiac muscle fibers of OA WT mice clarifying the presence of numerous severely damaged hypertrophied vacuolated mitochondria with completely destructed cristae (asterisk). (**I**,**J**) Electron micrographs of cardiac muscle fibers of OA + aMT WT mice exhibiting the beneficial effect of melatonin supplementation in keeping normal mitochondrial architecture (M). (**A**,**C**,**E**,**G**,**I**): bar = 2 μm and (**B**,**D**,**F**,**H**,**J**): bar = 1 μm.

**Figure 7 antioxidants-09-01187-f007:**
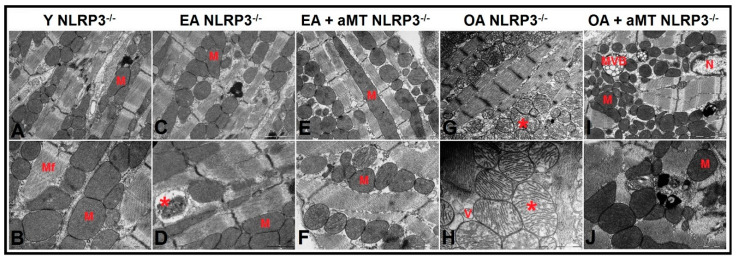
Age-related ultrastructural changes of mitochondria in cardiac muscle fibers of NLRP3^−/−^ mice and melatonin treatment. (**A**,**B**) Electron micrographs of cardiac muscle fibers of Y NLRP3^−/−^ mice showing the presence of normally highly compacted mitochondria with densely packed cristae (M) distributed among myofibrils (Mf). (**C**,**D**) Electron micrographs of cardiac muscle fibers of EA NLRP3^−/−^ mice demonstrating intact mitochondria (M) with individual ones depicting damaged peripherally cristae (asterisk). (**E**,**F**) Electron micrographs of cardiac muscle fibers of EA + aMT NLRP3^−/−^ mice revealing the clearly apparent prophylactic effect of melatonin supplementation in keeping normal mitochondrial architecture (M) with aging. (**G**,**H**) Electron micrographs of cardiac muscle fibers of OA NLRP3^−/−^ mice indicating less detectable mitochondrial damage compared with WT mice, with the presence of numerous mitochondria showing widely-separated organized cristae (asterisk) and small-sized membranous vacuoles of possibly autophagic nature (V). (**I**,**J**) Electron micrographs of cardiac muscle fibers of OA + aMT NLRP3^−/−^ mice showing the protective effect of melatonin supplementation in preserving normal mitochondrial structure (M), with formation of multivesicular bodies (MVB), which reflect the induction of the autophagic processes, N; nucleus. (**A**,**C**,**E**,**G**,**I**): bar = 2 μm and (**B**,**D**,**F**,**H**,**J**): bar = 1 μm.

**Figure 8 antioxidants-09-01187-f008:**
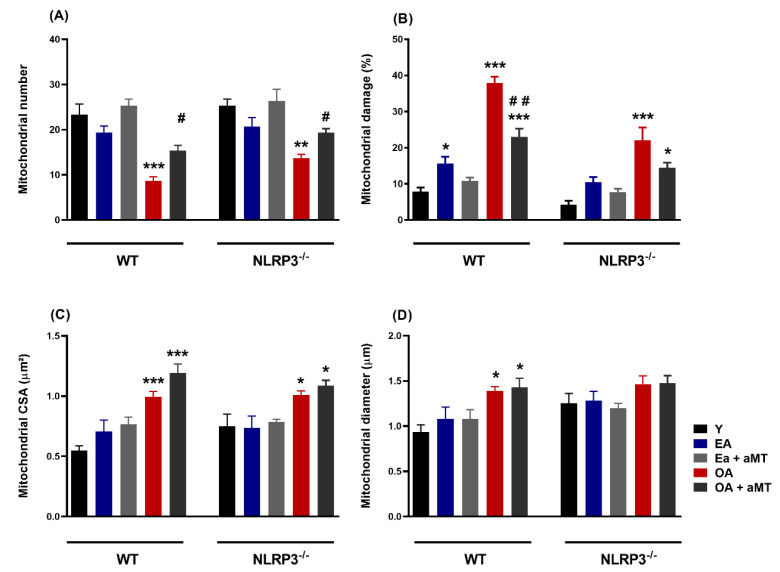
Age-associated morphometrical changes of intermyofibrillar mitochondria in cardiac muscle fibers of WT and NLRP3^−/−^ mice and melatonin treatment. (**A**) Analysis of mitochondrial number. (**B**) Analysis of mitochondrial damage percentage. (**C**) Analysis of cross-section area (CSA, µm^2^). (**D**) Analysis of mitochondrial Feret’s diameter (µm). Data are expressed as means ± SEM (*n* = 7 animals/group). * *p* < 0.05, ** *p* < 0.01, *** *p* < 0.001 vs. Y; ^#^
*p* < 0.05, ^##^
*p* < 0.01 vs. group without melatonin treatment.

**Figure 9 antioxidants-09-01187-f009:**
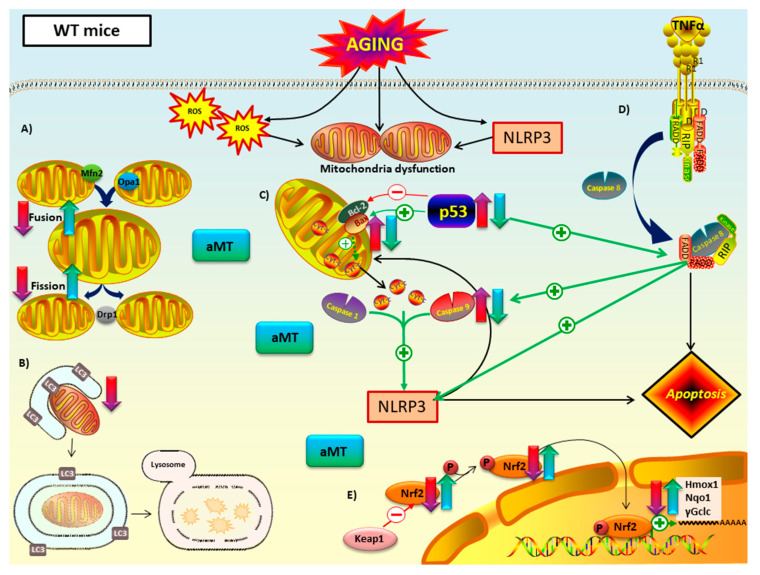
Proposed mechanism of melatonin in mitochondria of WT mice during cardiac aging. (**A**) Mitochondrial dynamics: aging led to a decline in fusion (Mfn2 and Opa1) and fission proteins (Opa1). Melatonin treatment counteracted this decrease. (**B**) Autophagy (mitophagy): autophagic capacity dropped in aged myocardium. Melatonin therapy had minimal impact on this pathway. (**C**) Intrinsic and (**D**) extrinsic apoptosis: WT mice have intrinsic and extrinsic pathways mediated by p53 and caspase 9. Those apoptotic markers, as well as Bax/Bcl2 ratio, increased with aging and are related with NLRP3 activation. This inflammasome seemed to have a regulatory effect on the intrinsic apoptotic pathway, which depends on mitochondria cytochrome c release. Melatonin supplementation had an anti-apoptotic effect in both intrinsic and extrinsic apoptosis. (**E**) Nrf2-dependent antioxidant response: Nrf2 and pNrf2 (Ser40) were reduced with aging. This loss was linked to the decrease of the cytoprotective enzyme transcriptionally regulated by Nrf2: Hmox1, Nqo1 and γGclc. Melatonin recovered this antioxidant pathway. No changes in Keap1 were reported. Red-purple arrow: impact of aging. Green-blue arrow: effect of melatonin treatment.

**Figure 10 antioxidants-09-01187-f010:**
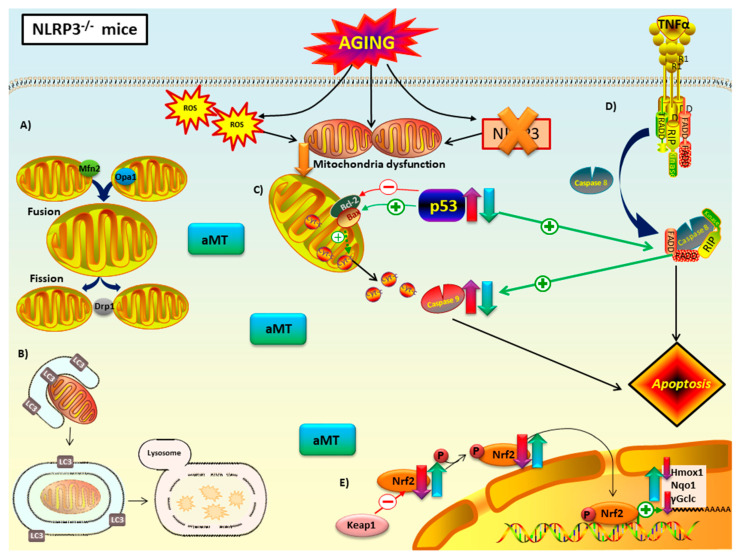
Proposed mechanism of melatonin in mitochondria of NLRP3^−/−^ mice during cardiac aging. Lack of NLRP3 inflammasome reduced mitochondria dysfunction. (**A**) Mitochondrial dynamics: the absence of NLRP3 prevented the decline in fusion (Mfn2 and Opa1) and fission proteins (Opa1) with aging. Melatonin treatment had no effect on these mice. (**B**) Autophagy (mitophagy): autophagic capacity was restored by NLRP3 deficiency. Melatonin therapy had minimal impact on autophagic capacity. (**C**) Intrinsic and (**D**) extrinsic apoptosis: loss of NLRP3 had an anti-apoptotic effect in Bax/Blc2 ratio, but not in p53 or caspase 9. The ablation of this inflammasome could trigger extrinsic apoptosis mediated by TNFα binding to death receptor. Melatonin supplementation had an anti-apoptotic effect in p53 and caspase 9. (**E**) Nrf2-dependent antioxidant response: lack of NLRP3 did not recover the decrease of this antioxidant pathway with aging. Only Nqo1 were not diminished in mutant mice. Melatonin improved this antioxidant pathway. No changes in Keap1 were reported. Red-purple arrow: impact of aging. Green-blue arrow: effect of melatonin treatment.

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
