# Peer review of "Melatonin/Nrf2/NLRP3 Connection in Mouse Heart Mitochondria during Aging"

_antioxidants, 2020, doi:10.3390/antiox9121187_

Round 1

Reviewer 1 Report

General comments:

This manuscript describes experiments characterizing a connection between melatonin and the NLRP3 inflammasome in the aging mouse heart.  Overall, the work is interesting and the experiments appear to be carefully performed and thoughtfully interpreted.  The authors may want to consider reducing the length of the “Discussion”.

Specific comments:

Abstract and Introduction:  Please state a hypothesis.

Page 3, line 109:  “sacrificed” – “killed” or “terminated” are better.

Page 3, line 113:  There needs to be more clarity here.  In the previous paragraph, it is stated “that hearts were collected , washed in saline and rapidly fixed”.  If that’s the case, where did heart tissue come from for Western blot analysis?

Figures:  Most journals and granting agencies are using scatter plots instead of bar graphs.  This practice does not change anything but is used for transparency.

Page 5, line 197:  “mechanism” should be “mechanisms”.

Page 6, line 217:  “no” should be “not”.

Page 7, line 245:  “few ones” should be “a few”.

Author Response

Amendments to the questions rose by the reviewer #1 to the manuscript ID antioxidants-999957 

General comments:

This manuscript describes experiments characterizing a connection between melatonin and the NLRP3 inflammasome in the aging mouse heart.  Overall, the work is interesting and the experiments appear to be carefully performed and thoughtfully interpreted.  The authors may want to consider reducing the length of the “Discussion”.

We are thankful for the general comments of the reviewer. The complexity and extent of the molecular pathways that have been addressed in this study required extensive detail in our discussion. Therefore, slight reductions were done.

Specific comments:

1) Abstract and Introduction:  Please state a hypothesis.

This comment has been addressed and a hypothesis has been added in the Abstract (lines 17-21) and Introduction (lines 73-81) sections.

2) Page 3, line 109:  “sacrificed” – “killed” or “terminated” are better.

It has been modified accordingly (page 3, line 115).

3) Page 3, line 113:  There needs to be more clarity here.  In the previous paragraph, it is stated “that hearts were collected , washed in saline and rapidly fixed”.  If that’s the case, where did heart tissue come from for Western blot analysis?

This comment was highly appreciated. We have revised the text to hopefully clarify this statement, as follows (lines 116-118):

Animals were killed by cervical dislocation after ketamine plus xylazine anesthesia, and hearts were collected. The left ventricle was dissected and divided into two parts. One part was washed in saline, and rapidly fixed in 2.5% glutaraldehyde for transmission electron microscopy analysis, while the other part was stored at -80ºC for further western blot analysis.

4) Figures: Most journals and granting agencies are using scatter plots instead of bar graphs. This practice does not change anything but is used for transparency.

We are aware of the importance of data transparency and we agree with the reviewer. However, we found more preferable to use graph bars. The high number of western blot bands made the graph bars more intuitive, understandable and clear in comparison to scatter plots.

5) Page 5, line 197:  “mechanism” should be “mechanisms”.

It has been revised as requested; it corresponds to page 5, line 206.

6) Page 6, line 217:  “no” should be “not”.

It has been revised as requested; it corresponds to page 6, line 226.

7) Page 7, line 245:  “few ones” should be “a few”.

It has been revised as requested; it corresponds to page 7, line 254.

Reviewer 2 Report

I read with interest the article:

“Melatonin/Nrf2/NLRP3 connection in mouse heart 2 mitochondria during aging”

submitted to Antioxidants Journal.

In my opinion this paper is acceptable in Antioxidants Journal with a minor revision, as below described.

Results, §3.2 (lanes 180-194, Figure 3):

  • The immunoblotting experiments in Fig. 3 (a, b) should be differently showed. In my opinion, the bands of lane LC3I and LC3II should be showed together in one image, because obtained by the same immunoblotting. This help to understanding the conversion of LC3I into LC3II and hence the regulation of autophagic process.
  • Did the authors tested different possible housekeeping protein to use them for the normalization of immunoblotting signals? The reason(s) of choosing GAPDH should be explained.

Materials and Methods § 2.5. Statistical analyses (lanes 157-161)

- Data were statistically analyzed using multiple comparison of the ANOVA test. Did the authors verify that the assumptions to apply the ANOVA test were fulfilled and which tests were used to test these hypotheses?

Author Response

Amendments to the questions rose by the reviewer #2 to the manuscript ID antioxidants-999957 

Comments and Suggestions for Authors:

I read with interest the article: “Melatonin/Nrf2/NLRP3 connection in mouse heart 2 mitochondria during aging” submitted to Antioxidants Journal. In my opinion this paper is acceptable in Antioxidants Journal with a minor revision, as below described.

1) Results, §3.2 (lanes 180-194, Figure 3): The immunoblotting experiments in Fig. 3 (a, b) should be differently showed. In my opinion, the bands of lane LC3I and LC3II should be showed together in one image, because obtained by the same immunoblotting. This help to understanding the conversion of LC3I into LC3II and hence the regulation of autophagic process.

We follow the suggestion of the reviewer. Bands of lane LC3I and LC3II are showed together in one image (Figure 3).

2) Did the authors tested different possible housekeeping protein to use them for the normalization of immunoblotting signals? The reason(s) of choosing GAPDH should be explained.

We did not test different possible housekeeping protein to use for the normalization of immunoblotting signals. However, before performing the experiments, we investigated the most used loading controls for cytoplasmic proteins, which were actin (43 kDa), GAPDH (37 KDa) and tubulin (55 kDa). We finally chose GAPDH because: 1) its molecular weight was different from our targets proteins; 2) its expression remains stable with aging [1,2]. A recent Nature report established that the stability of GAPDH with aging is sufficient to support its extended use as a housekeeper [3]. We therefore concluded that this housekeeping gene is suitable and reliable for normalization of our immunoblots.

  1. Barber, R.D.; Harmer, D.W.; Coleman, R.A.; Clark, B.J. GAPDH as a housekeeping gene: analysis of GAPDH mRNA expression in a panel of 72 human tissues. Physiol. Genomics 2005, 21, 389–395, doi:10.1152/physiolgenomics.00025.2005.
  2. Touchberry, C.D.; Wacker, M.J.; Richmond, S.R.; Whitman, S.A.; Godard, M.P. Age-Related Changes in Relative Expression of Real-Time PCR Housekeeping Genes in Human Skeletal Muscle. J. Biomol. Tech. JBT 2006, 17, 157–162.
  3. González-Bermúdez, L.; Anglada, T.; Genescà, A.; Martín, M.; Terradas, M. Identification of reference genes for RT-qPCR data normalisation in aging studies. Sci Rep 2019, 9, 13970, doi: 10.1038/s41598-019-50035-0.

3) Materials and Methods § 2.5. Statistical analyses (lanes 157-161)

- Data were statistically analyzed using multiple comparison of the ANOVA test. Did the authors verify that the assumptions to apply the ANOVA test were fulfilled and which tests were used to test these hypotheses?

To perform the ANOVA test, we initially assume Gaussian Distribution of residuals and similar sample variation. These hypotheses were assessed using the Shapiro-Wilk normality test and the Brown-Forsythe test, respectively.